# The Long-Range Biomimetic Covert Communication Method Mimicking Large Whale

**DOI:** 10.3390/s22208011

**Published:** 2022-10-20

**Authors:** Jongmin Ahn, Deawon Do, Wanjin Kim

**Affiliations:** Agency of Defense Development, Changwon-si 516852, Korea

**Keywords:** underwater communication, biomimicking communication, covert communication

## Abstract

Short-range biomimetic covert communications have been developed using dolphin whistles for underwater acoustic covert communications. Due to a channel characteristics difference by range, the conventional short-range methods cannot be directly applied to long-range communications. To enable long-range biomimicking communication, overcoming the large multipath delay and a high degree of mimic (DoM) in the low-frequency band is required. This paper proposes a novel biomimetic communication method that preserves a low bit-error rate (BER) with a large DoM in the low-frequency band. For the transmission, the proposed method utilizes the time-dependent frequency change of the whistle, and its receiver obtains additional SNR gain from the multipath delay. Computer simulations and practical ocean experiments were executed to demonstrate that the BER performance of the proposed method is better than the conventional methods. For the DoM assessment, the novel machine learning-based method was utilized, and the result shows that the whistles generated by the proposed method were recognized as the actual whistle of the right humpback whale.

## 1. Introduction

Recently, a biomimetic communication method has been studied for military purposes. Biomimetic communication methods make the enemy confuse our communication signals with dolphin or whale sounds to achieve covertness [1,2,3,4,5,6,7,8,9,10,11,12,13,14]. Traditional covert communication systems decrease the signal strength and hide the existence of the signal using various techniques (e.g., spread spectrum, chaotic modulation, etc.) [15,16,17,18]. Low-power spectral density of the transmitted signal causes a low SNR at the receiver, which results in low bit-error-rate (BER) performance. Especially for long-distance communication, low frequency and narrow bandwidth are inevitable, and it is difficult to conceal the signal using these methods [7,8,9,10,11,12,13,14]. Biomimetic underwater acoustic communication has been developed to overcome the disadvantage of traditional covert communication. This method allows high transmit power without considering a low detection probability, and large SNR at the receiver can be preserved.

Conventional biomimetic communication methods utilize various modulation methods such as chirp spread spectrum (CSS), frequency shift keying (FSK), phase shift keying (PSK), and time-frequency shift keying (TFSK) to modulate bits into whistles [7,8,9,10,11,12,13,14]. Conventional biomimetic communication studies have demonstrated their communication performance through ocean experiments. In addition, the conventional methods have evaluated the degree of mimic (DoM) by comparing the shape of mimicking the signal with the real whistle. Especially in studies of CV-CFM (based on PSK) and TFSK, the authors performed the mean-of-score (MOS) test and showed that humans could hardly distinguish a mimicking signal from a real dolphin whistle [11,12,13,14]. All of the conventional studies mimicked the whistle of small dolphins (such as white-sided dolphins utilizing tens of kilohertz sounds) and conducted ocean experiments at a short range within 5 km [1,2,3,4,5,6,7,8,9,10,11,12,13,14]. However, considering the operation area of military-purpose underwater vehicles from tens to hundreds of km, biomimetic communication should support long ranges. For long-distance communication, using a low-frequency signal (i.e., less than several hundred hertz) is advantageous over a high-frequency signal (i.e., over tens of kilohertz) due to absorption loss, and large whales use these low-frequency whistles. Thus, it is necessary to mimic a large whale for long-range biomimicking communication, and long-range underwater acoustic communication is difficult with conventional biomimetic methods.

The first problem is an increase in BER due to a large multipath delay. As the range between transmitter and receiver increases to tens of kilometers, the multipath delay also increases up to a few seconds [19,20,21,22,23,24,25]. However, conventional biomimetic communication methods do not consider a few seconds of multipath delay because these methods are designed for a short communication range within 5 km. Thus, when a multipath delay occurs for a few seconds, the BER of the conventional biomimetic methods increases. The second problem is a degradation of the DoM performance. Human psychoacoustic ability is the most sensitive to sound in hundreds of hertz to kilohertz [26]. On the other hand, small dolphins utilize high-frequency whistles exceeding the human auditory sense. When the conventional methods mimic small dolphins, these facts have an advantage in the DoM performance. As a result of the DoM evaluation in Ref. [14], it was determined that the DoM of killer whales (2 kHz~4 kHz) is lower than that of white-sided dolphins (15 kHz~25 kHz). Therefore, two things are required to enable long-range communication with biomimetic modulation. The first one is the robustness against large multipath delay, and the second is the performance improvement of the DoM for whistles with hundreds of hertz to kilohertz. In this paper, we propose a novel long-range biomimetic modulation method that has robustness against large multipath delay and higher DoM performance than the conventional methods.

Since the whistle is a non-linear chirp signal, it has strong robustness against the distortion of large multipath delay. Thus, if the bit can be modulated while maintaining the shape of the whistle, the modulated signal has tolerance to the large multipath delay and the high DoM performance. Whales generate whistles with various frequency contours, but they have a habit when generating the sounds of the whistle. The whistle-generating habit is that the following frequency values are limited by the frequency value of the previous time. The proposed biomimicking modulation method utilizes this frequency-change habit of the whistle.

The main contributions of the paper are summarized as follows:We modeled the time-dependent frequency change of the whistle of the right humpback whale (RHW) using the Markov chain (MC). The modeled results can be employed to select the frequency values used by the whistle. Note that we considered the time-frequency orthogonality of the symbol to preserve low BER when modeling the frequency change of the whistle.The proposed modulation method preserves the signal shape of the whistle because it modulates bits using frequency values that whales use to generate a whistle. Thus, the whistle-mimicking communication signal generated by the proposed modulation method shows improved performance of the DoM and tolerance of large multipath delay.The proposed demodulation method estimates the multipath delay profile using a preamble and increases received signal gain using the estimated profile. Thus, the proposed method achieves low BER in a long-range underwater environment with a large multipath delay.Computer simulations and practical ocean experiments were conducted and demonstrated that the proposed method had a lower BER than conventional covert communication methods.Since the proposed modulation method makes a whistle using modeling results, a machine learning-based DoM assessment was conducted. The assessment results show that the trained machine learning classifier recognized the whistle-mimicking signal generated by the proposed modulation method as the whistle of the RHW. This result shows that machine learning can be used as an effective evaluation method for the DoM performance of biomimetic communication.

This paper is organized as follows. Section 2 describes how to model the time-dependent frequency change of the whistle of the RHW using MC. In Section 3, the proposed biomimicking modulation and demodulation methods are described. Section 4 shows the result of a machine learning-based DoM assessment. In Section 5, through computer simulations and ocean experiments, it is shown that the proposed biomimicking communication method has a lower BER than conventional methods. Section 6 concludes the paper.

## 2. Modeling Whistles of the Right Humpback Whale

Whales communicate with each other using whistles. For the long-range biomimicking communication method, we selected the RHW as a mimicking model. The RHW is about 15 m long and makes sounds from tens to hundreds of hertz. Figure 1 shows the spectrogram of the whistles of the RHW. 

Figure 1 shows three frequency contours of the whale whistle starting at 150 Hz. In Figure 1a,d, the frequency of the whistle is 150 Hz until halfway through the whistle. Then it increases to 250~300 Hz. In Figure 1b,e, the whistle frequency increases linearly from 150 Hz to 300 Hz. The whistles in Figure 1c,f also start at 150 Hz but end at 200 Hz. As shown in Figure 1, whales generate various whistles, but we can see that there is a specific habit when whales make whistle sounds. The frequency values of the whistle are limited according to their habits [27,28,29,30]. For example, if the RHW starts the whistle at 150 Hz, we can infer that whales are used to generating sounds under 300 Hz halfway through the whistle. In this paper, we propose a biomimicking communication method using the whistle-generating habit of whales.

One of the simple examples of the proposed method is shown in Figure 1d,e. The whistles in Figure 1d,e start at 150 Hz. Then, at 0.55 s of whistle time, bit 0 is modulated when the frequency value of the whistle is 150 Hz (red box). If the frequency value of the whistle is 200 Hz, it indicates bit 1 (white box). Likewise, if the whistle-generating habit of the RHW is modeled, we can transmit binary information according to the modeling results. As shown in Figure 1, pre-recorded whistles are already contaminated with noise, and communication performance cannot be guaranteed when the pre-recorded whistle is utilized directly. Therefore, in this paper, a whale whistle is modeled.

The whistle-generating habit of the whale is that the subsequent frequency values are restricted by the frequency value of the previous time [27,28,29,30]. This habit is suitable for Markov chain (MC)-based modeling, and some studies have analyzed the time-dependent frequency change of whistles using MC [31,32,33,34,35,36,37]. However, applying conventional modeling methods directly to the proposed methods is difficult due to the degradation of BER performance. The time-frequency orthogonality between symbols must be maintained for the low BER. However, conventional modeling methods do not consider this orthogonality. Therefore, the time-dependent frequency change of the whistles is modeled in a time-frequency unit that satisfies the orthogonality. The proposed method represents information bits using the frequency change with time. The symbol can be distinguished when the time and frequency change is greater than twice the reciprocal of the symbol duration, such as linear frequency modulation (LFM). 

For the modeling, the frequency values of a whistle over time were extracted by short-time Fourier transform (STFT). When the time window length of STFT is ΔT, the minimum frequency unit that guarantees the orthogonality between two different LFMs is 2/ΔT. Thus, the frequency resolution of STFT ΔF is 2/ΔT. As a result of STFT, the whistle is represented in the 2D domain with K-length of the time sample and L-length of the frequency sample. Figure 2a shows the (L×K) size of the 2D whistle. In Figure 2,$\;T_{w}$ is the time length of the whistle. We can obtain one frequency value for each K column from the 2D whistle, and the time-dependent frequency change is obtained as $F=\left[f_{1},\ldots ,f_{k},\ldots ,f_{K}\right]$. If we obtain the F matrix from N whistles, the probability distribution of the frequency change over time can be obtained using Markov-chain modeling. 

It is assumed that there are M whistles with a frequency value of ΔF×l at an arbitrary time ΔT×k. Among these M whistles, only $M^{\prime }$ whistles have the increased frequency value ΔF×(l+1) at the next time ΔT×(k+1). When the frequency value of the whistle is ΔF×l at ΔT×k, the probability that the frequency value is ΔF×(l+1) in the next time interval ΔT×(k+1) becomes $M^{\prime }/M$. This conditional probability can be expressed as $P\left(\;f_{k}=\Delta F\times l|\;f_{k+1}=\Delta F\times \left(l+1\right)\right)$. If this conditional probability is 0, the whale does not use the whistle pattern that changes from ΔF×l to ΔF×(l+1). By applying the Markov chain to the calculated conditional probabilities, we can model the time-dependent frequency-change rule of the whistle of the RHW. As a result of modeling, when the frequency value is $\;f_{k-1}$ at ΔT×k, the set of the frequency values generated by the whale at ΔT×(k+1) is expressed as Equation (1):(1)$$F_{k+1}=\left\{\;\Delta F\times l\;|\mathrm{Pr}\left(\;f_{k}|\;f_{k+1}=\Delta F\times l\right)>\mathrm{Thr},\;l=1\sim L,\;f_{k}\in F_{k}\;\right\},\;k=1\sim K.$$

In Equation (1), $F_{k+1}$ is the set of frequency values used by a whale at ΔT×(k+1). Thr is a threshold value for selecting only frequency values usually generated by whales, and we set it as 0.5. Note that frequency values in $F_{k+1}$ are quantized by ΔF, which guarantees symbol orthogonality. On the other hand, the actual whale whistle has time-frequency ambiguity, as shown in Figure 1. The time-frequency ambiguity of the whistle is also considered by modeling the time-dependent frequency change of the whistle with various lengths of the STFT window. Figure 2b shows the STFT results with another window length. 

In Figure 2, ΔT is smaller than $\Delta \hat{T}$. For example, let us assume that $\Delta \hat{T}$ (e.g., $\Delta \hat{T}$: 0.2) is 2ΔT (e.g., ΔT: 0.1). The time resolution of Figure 2a is higher than that of Figure 2b. However, the frequency resolution of Figure 2a is lower than that of Figure 2b. As shown in Figure 2, the time-frequency ambiguity of the whistle is considered by conducting STFT with various lengths of the window. In this paper, we modeled the whistle-generating habit using 6700 whistles of the RHW [38] and six lengths of the STFT window (i.e., 0.075 s, 0.1 s, 0.125 s, 0.15 s, 0.175 s, 0.2 s). Figure 3 shows some examples of the modeling results.

In Figure 3, blue lines show the possible variation of whistle frequency. As shown in Figure 3, whistle lengths are quantized to the mean value of each time length of $T_{w}$. Since ΔT of Figure 3a is shorter than that of Figure 3b, whistles are divided into three parts in (a) and two parts in (b). In Figure 3, the RHWs are used to start within 130~165 Hz when generating a whistle. Thus, in Figure 3b, starting frequency values ($F_{1}$) are quantized as (135 Hz, 145 Hz, 155 Hz) and frequency values at the half-time of the whistle ($F_{2}$) are quantized as (145 Hz, 155 Hz, 165 Hz, 175 Hz, 185 Hz). As a result of the model in Figure 3b, the RHW does not generate exceeding 165 Hz at half of the whistle when the start frequency of the whistle is 135 Hz. The proposed modulation method utilizes this whistle-generating habit of the RHW to modulate bits. A simple example is represented in Figure 3b. When the whistle starts at 135 Hz, there are two available frequency values (145 Hz and 155 Hz) at 0.2 s. In this case, 145 Hz represents 1 bit and 155 Hz is 0 bits. In Figure 3b, the given parameters satisfy the time-frequency orthogonality condition (ΔF≥2/ΔT) and low BER can be achieved. Section 3 describes the proposed method in detail.

## 3. The Proposed Biomimicking Communication Method

The proposed method utilizes the model of the whistle-generating habit of the RHW when it modulates/de-modulates bits. The transmitter and receiver share these modeling results. The proposed transmitter uses a pseudo-random number generator and randomly selects one model among the models with various lengths of the window. It is assumed that the receiver also selects the same model using the same pseudo-random number generator. Firstly, the proposed biomimicking modulation method is described.

### 3.1. The Proposed Biomimicking Modulation Method

The time-frequency units of the model (ΔT and ΔF) satisfy symbol orthogonality and become the symbol time frequency. The whistle is divided by the symbol length ΔT. Let us assume that whistle is divided into K symbols. Information bits are mapped to the available frequency values at each symbol period. Figure 4 shows an example. In the (k−1)–th symbol period, there are four available frequency values and two bits are modulated. If 10 is modulated to the (k−1)–th symbol, there are two available frequencies in the k–th symbol period. Thus, the  k–th symbol modulates one bit, whereas the (k−1)–th symbol modulates two bits. That is, the number of modulation bits in the k–th symbol depends on the frequency value selected in the (k−1)–th symbol. $M_{k}$ is the modulation order in the k–th symbol. 

In Figure 4, the proposed method selects the available whistle frequency according to the bits using gray coding and generates the entire biomimicking whistle using the selected frequency. The modulation process is described in detail. When k is 1, available frequency values are elements of the set $F_{1}$ in Equation (1) and the modulation order $M_{1}$ is calculated as $log_{2}\left(\left|F_{1}\right|\right)$. Thus, $M_{1}$ bits are transmitted with the first symbol. Assume that $B_{M_{1}}$ is the set of all $M_{1}$ bit sequences. Each element of $B_{M_{1}}$ is mapped one by one to an element in a set $F_{1}$ using gray coding. When all the bit sequences in $B_{M_{1}}$ are converted to decimals, the converted result is a set of integers from 0 to $2^{M_{1}}-1$. This set of integers becomes the symbol set ($S_{1}$) for the first symbol and each element of $S_{1}$ is an index of $F_{1}$. Let us assume that $b_{1}$ is the Tx bits sequence and it is converted to the symbol $s_{1}$($s_{1}\in S_{1}$). The whistle frequency of the first symbol is selected with $F_{1}\left(s_{1}\right)$. When k is 2 or more, the available frequency in the k-th symbol depends on the selected frequency in the (k−1)-th symbol. Therefore, all frequencies in the set $F_{k}$ cannot be used, as shown in Figure 4. When $G_{k}$ is the set of frequency values available in the k-th symbol and $G_{k}$ is expressed by Equation (2), then


(2)
$$\left\{ \begin{array}{cc} k\geq 2, & G_{k}=\left\{\Delta F\times l|\mathrm{Pr}\left(G_{k-1}\left(s_{k-1}\right)=f_{k-1}|f_{k}=\Delta F\times l\right)>Thr,l=1\sim L\right\}\\ k=1, & G_{1}=F_{1} \end{array}.\right.$$


In Equation (2), when k is 1, any frequency in the set $F_{1}$ is available. Thus, $G_{1}$ is the same as $F_{1}$. If k is greater than or equal to 2, a frequency value is selected from the set $G_{k}$ according to the Tx bits $b_{k}$ and the modulation order $M_{k}$ is calculated as $log_{2}\left(\left|G_{k}\right|\right)$. Therefore, likewise, when k is 1, the whistle frequency for the k-th symbol is selected as $G_{k}\left(s_{k}\right)$ and the biomimicking communication signal in the k-th symbol period is expressed by Equation (3):(3)$$W_{k}\left(t\right)=\cos(2\pi (\frac{\left(G_{k-1}\left(s_{k-1}\right))-G_{k}\left(s_{k}\right)\right)}{2\Delta T}t^{2}+G_{k-1}\left(s_{k-1}\right)t)).$$

In Equation (3), the biomimicking communication signal in the k-th symbol period is LFM, as shown in Figure 4. The entire biomimicking communication signal is the sum of all the signals from 1 to the K-th symbol period, and it is obtained by Equation (4):(4)$$W\left(t\right)=\sum_{k=2}^{K}W_{k}\left(t-\Delta Tk\right).$$

The block diagram of the proposed biomimicking modulation is shown in Figure 5.

A preamble is inserted in front of the biomimicking communication signal to find the received signal and estimate the channel delay at the receiver. This preamble is the original whistle of the whale. Let us assume that the receiver also has the same preamble. The frame structure of the biomimicking communication signal is shown in Figure 6.

The proposed biomimicking whistle is the summation of several LFMs. Therefore, like LFM, the auto-correlation function becomes a Kronecker delta function. If the channel delay can be estimated, an additional signal gain can be obtained using signals received through multiple paths, reducing bit error. The following section describes the reception method.

### 3.2. The Proposed Biomimicking Demodulation Method

The transmitted signals generated using Equation (4) pass through the underwater channel and are received at the receiver. The received signal y(t) is modeled as
(5)$$y\left(t\right)=\sum_{i=1}^{I}H_{i}\times W\left(t-\tau_{i}\right)+n\left(t\right).$$

In Equation (5), $H_{i}$ is the channel gain of the i-th path and n(t) denotes an additive white Gaussian noise (AWGN). The number of the path is I. The receiver generally finds the received signal on the path with the greatest channel gain using the preamble. Thus, we assume that $\tau_{1}$ is zero and $H_{i}$ has the maximum value when i is 1. For the precise detection of received bits, a maximum likelihood sequence detection (MLSD)-based receiver is proposed. In Equation (2), the k-th symbol is modulated depending on the (k−1)-th symbol. Therefore, ML detecting for the sequence with K symbols shows a lower error rate than detecting each of the K symbols as a single one. When the sequence of the transmitted K symbols is $S_{\mathrm{Tx}}^{*}=\left\{s_{1}^{*},\ldots ,s_{k}^{*},\ldots ,s_{K}^{*}\right\}$, detecting the sequence of symbols with the maximum likelihood probability for $S_{\mathrm{Tx}}^{*}$ is expressed as (6)$$\begin{array}{cc} {\hat{S}}_{\mathrm{Rx}} & =\underset{S_{1},\ldots ,S_{K}}{\max}P\left({\;G}_{1}\left(s_{1}^{*}\right)={\;G}_{1}\left(s_{1}\right),\ldots ,G_{k}\left(s_{k}^{*}\right)=G_{k}\left(s_{K}\right),\ldots ,G_{K}\left(s_{K}^{*}\right)={\;G}_{K}\left(s_{K}\right)\right)\\ & =\underset{S_{1},\ldots ,S_{K}}{\max}\prod_{k=2}^{K}P\left({\;G}_{k-1}\left(s_{k-1}^{*}\right)={\;G}_{k-1}\left(s_{k-1}\right)|{\;G}_{k}\left(s_{k}^{*}\right)={\;G}_{k}\left(s_{k}\right)\right) \end{array}.$$


To find a symbol sequence that satisfies Equation (6), the proposed demodulation method calculates a correlation between the transmitted whistle and all whistles that the model can generate. When the generated whistle is $\hat{W}\left(t\right)$ with the given model and K symbols {$s_{1},\ldots ,s_{K}$}, the correlation between the generated whistle $\hat{W}\left(t\right)$ and transmitted whistle W(t) is expressed as
(7)$$\begin{array}{cc} R\left(\tau \right) & =\int^{\;}W\left(t-\tau \right)\times \hat{W}\left(t\right)dt\\ & =\sum_{k=2}^{K}\int^{\;}W_{k}\left(t\right)\times {\hat{W}}_{k}\left(t\right)dt\;\\ & =\sum_{k=2}^{K}R_{k}\left(\tau \right) \end{array}.$$

As derived from Equation (7), the correlation between $\hat{W}\left(t\right)$ and W(t) is equal to the summation of K symbol correlation values $R_{k}\left(\tau \right)$ that are calculated with the transmitted and generated whistle for each symbol. Therefore, if the symbol sequence satisfies Equation (6), the signal generated by the corresponding sequence has the maximum correlation with the received signal. The correlation between y(t) and $\hat{W}\left(t\right)$ is expressed as Equation (8).
(8)$$y\left(t\right)*\hat{W}\left(t\right)=\sum_{i=1}^{I}H_{i}\times R\left(\tau -\tau_{i}\right)+n^{\prime }\left(\tau \right),$$
where ∗  denotes a correlation operation. As mentioned above, when ${\hat{W}}_{k}\left(t\right)$ is equal to $W_{k}\left(t\right)$, R(τ) is Kronecker delta function δ(τ)  and the result of Equation (8) is the same as the delay profile. Note that $H_{i}$ has the maximum value when i is 1. Thus, the maximum value of Equation (8) is the gain of the first path $H_{1}$, whereas the proposed demodulation method obtains an additional signal gain from the received signal through the other paths. Let us assume that the preamble has gone through the same multipath as the received signal. The estimated channel using the preamble is $\sum_{i=1}^{I}\;H_{i}\times \delta \left(\tau -\tau_{i}\right)$. To obtain the additional signal gain, integrate and dump the estimated channel and Equation (8). The result of the integration and dump is expressed as follows:(9)$$\{\begin{array}{l} H0\left(W_{k}\left(t\right)={\hat{W}}_{k}\left(t\right)\right):P=\sum_{i=1}^{I}H_{l}{}^{2}+N^{\prime }\\ H1\left(W_{k}\left(t\right)\neq {\hat{W}}_{k}\left(t\right)\right):P=\sum_{i=1}^{I}H_{l}{}^{2}\times \int^{\;}R\left(\tau \right)d\tau +N^{\prime } \end{array}.$$

If ${\hat{W}}_{k}\left(t\right)$ is equal to $W_{k}\left(t\right),$ Equation (9) is the summation of I path gain, whereas when ${\hat{W}}_{k}\left(t\right)$ and $W_{k}\left(t\right)$ are different, ∫R(τ)dτ has a very small value because R(τ) becomes the cross-correlation of different LFM signals. Therefore, Equation (9) becomes the maximum value when ${\hat{W}}_{k}\left(t\right)$ is equal to $W_{k}\left(t\right)$ and the received symbol sequence is detected as Equation (10):(10)$${\hat{S}}_{\mathrm{Rx}}=\underset{S_{1},\ldots ,S_{K}}{\max}P.$$

The block diagram of the proposed biomimicking demodulation method is depicted in Figure 7. 

## 4. Evaluation of Degree-of-Mimic Performance

This section evaluates the degree-of-mimic (DoM) performance of the proposed method. Quantitative (i.e., spectral correlation) and qualitative methods (i.e., MOS-test) have been proposed to measure the degree of mimic in conventional biomimetic studies [8,9,10,11,12,13,14]. Spectral correlation measures the similarity of the time-frequency change between the actual whale whistle, and the MOS test evaluates whether a person recognizes a mimicking whistle as an actual whale whistle. The results of conventional studies show that the biomimicking whistle has the best score in both DOM evaluation methods (i.e., spectral correlation and MOS test) when it is not distorted during the modulation process [10,11]. Since the proposed method does not distort the whale whistle, it has the highest score of the two methods and is undetected by conventional communication signal-recognition techniques, which find the artificial and periodical signal characteristics. Therefore, this paper proposes a novel DoM evaluation method to more precisely evaluate the DoM of biomimicking signals.

The proposed DoM evaluation method discriminates even if the biomimicking whistle is the sound of a specific whale species. The proposed method not only judges whether a biomimicking whistle sounds like a whale, but also determines whether it simply sounds like a specific species (e.g., right humpback, killer, etc.). Determining a species of whale with only whistles is a professional task and requires much experience. Thus, to complete this task effectively, whistle classifiers based on machine learning have been studied [39,40]. In this paper, the machine-learning classifier was utilized to evaluate the DoM performance of the proposed method. The classifier that won the Marine Explore and Cornell University Whale Detection Challenge competition was used for evaluation [38]. To evaluate the DoM, 1000 biomimicking whistles were generated using the model in Section 3. These 1000 whistles were modulated with conventional methods (i.e., CV-CFM and TFSK [10,13] and FSK [6,7,8]) to compare the DoM performance of the proposed method with that of conventional methods. TFSK-modulated biomimicking whistles are mathematically the same signal as CV-CFM-modulated whistles [13]. Thus, these two modulation schemes shared whistles for evaluation. Note that all models with six window lengths were utilized to generate 1000 whistles. Figure 8 shows examples of the whistles generated for the DoM evaluation. 

The whistle in Figure 8 was generated by the model with a 0.2 s window length. The results of the DoM evaluation are shown in Table 1.

In Table 1, 95% of whistles generated by the proposed method were recognized as the whistle of the RHW. Note that the classifier did not simply identify the given sound as an actual whistle of a whale but rather as the whistle of the RHW. Thus, this result demonstrates that the proposed model generated a whistle like the real RHW whistle. However, in cases of FSK- and CV-CFM-modulated whistles, only 10% and 60% were recognized as the whistle of the RHW, respectively. Note that these three types of whistles have the same frequency contour; the only difference is the modulation scheme. Therefore, this result shows that the conventional methods distorted the whistle when modulating bits. This distortion decreased the DoM performance. In addition, as seen in Figure 8b,c, the existing method distorted the signal so humans could recognize this difference and the MOS score decreased. The value of spectral correlation also decreased. As a result of the DoM evaluation, the DoM performance of the proposed method is superior to that of the conventional methods. In the next section, the computer simulations and ocean-experiment results are shown for communication-performance comparisons of the proposed method with the conventional methods.

## 5. Evaluation of Communication Performance

### 5.1. Simulation

The BER performance of the proposed biomimicking method was compared with that of conventional methods (i.e., FSK, CV-CFM, and TFSK) in the long-range underwater environment. The modulation order and parameter were selected for the fair BER comparison to yield a similar data rate. These values are given in Table 2. M is the modulation order.

In Table 2, since all models with six window lengths were utilized for the proposed method, the data rate is an average value. In Table 2, let N be the number of transmitted whistles for simulation. $b_{n}$ is the number of bits transmitted by the n-th whistle and $T_{n}$ is the time length of this whistle. The data rate is obtained as ($\sum_{n=1}^{N}b_{n}/T_{n})/N$. The data rate of the conventional methods is about 12~15 bps, except for TFSK. The simulations were performed with the bandwidth used by the whale, and the data rate of the TFSK is limited.

There are many simulators for modeling underwater acoustic sound propagation. However, we referred to the measured channel from the experiments of long-range (i.e., longer than 50 km) underwater acoustic communication for realistic simulation. Table 3 summarizes the cases of the long-range underwater acoustic communication experiment conducted in the bandwidth of the RHW. The depiction of the channel measured in the experiments is also described in Table 3. 

In Table 3, the maximum channel delay is 2.5 s and the maximum number of the multipath is 17. Based on these facts, three channels were generated by the bellhop according to maximum delay (i.e., 30 ms, 1 s, and 2 s), and these three channels were utilized for simulation. The three channels used for simulation and the BER results are displayed in Figure 9.

In Figure 9, the orange line denotes the BERs of the proposed method. The blue line is the BER of the proposed method without multipath gain. The yellow, purple, and green indicate the BERs of the conventional biomimicking communication methods (i.e., CV-CFM, FSK, TFSK, respectively). In Figure 9, the proposed method showed better BER performances than the conventional algorithms. The conventional methods had error floors at 10^−2^~10^−1^. When SNR was lower than −10 dB, TFSK showed the lowest BER. However, the data rate of the TFSK was also half that of the others. When proposed without multipath gain, the BER performance of the proposed method was similar to that of TFSK. Please note that, as shown in Table 2, the data rate of the proposed method was higher than that of TFSK. Therefore, the BER performance of the proposed method was better even without multipath combining. In addition, increasing SNR did not decrease the BER of the TFSK. The proposed method decreased BER by taking multipath gain. These results show that it is difficult for conventional methods to overcome the large multipath delay in the long-range underwater communication environment. However, the proposed method obtained the additional SNR gain using signals received from multipath and showed the lowest BER with 1~2 s multipath delay, as depicted in Figure 9b,c. In the following subsection, the results of the practical ocean experiments are described.

### 5.2. Ocean Experiment

Practical ocean experiments with the proposed and conventional methods were executed 60 km away, eastbound from Pohang city, on 18 May 2022. The range between the transmitter and the receiver was over 60 km. The average depth of the ocean was about 1100 m. The transmitter was located at 295 m with a depth of 850 m. The bandwidth of the transmitter was 2.6 kHz to 4.6 kHz, and we upconverted the center frequency of the whistle to 3 kHz. The three-channel receiver was utilized, and each sensor was positioned at 205 m, 230 m, and 255 m below the ocean surface. The depth of the receiver was about 1200 m. Figure 10 shows the experiment location and the configuration of the experiments.

The parameters of the proposed and conventional methods were the same as those of computer simulations in Table 2. When bits are utilized as orthogonal code, CV-CFM-modulated whistles become TFSK-modulated biomimicking whistles [14]. Thus, these two modulation schemes shared whistles for evaluation. The DoM performance of the FSK biomimicking signal showed about 10%, which is too low to use practically. Thus, the FSK biomimicking method was not transmitted during the ocean experiment. For the BER performance calculation, 945 whistles were transmitted for each method. The proposed and conventional methods (i.e., CV-CFM and TFSK) were alternatively sent and went through the same underwater acoustic channel. Figure 11a,b depict the multipath delay profile of the proposed method and the conventional methods. 

In Figure 11, the same multipath delays were measured for each method. It can be seen that more than two paths occurred between 30 ms and 40 ms. The three sensors were 25 m apart from each other, and the depths of the three sensors were different. Therefore, the received signal of each sensor went through a different underwater communication channel, as shown in Figure 11. The uncoded BERs of each sensor are shown in Table 4.

The average BER of the proposed method was 0.0057, whereas the conventional method exhibited 0.079 and 0.1165. The BER of the proposed method without the multipath was 0.009, which is also lower than conventional methods. This result shows that the proposed method obtained gains from the multipath. In the ocean experiments, the SNR of the received signal was estimated at about 5 dB, and the multipath delays of the second and third sensors were similar to those of the simulation experiment in Figure 9a. Therefore, the BERs of those two sensors were similar to the BER in Figure 9a. As shown in Figure 11, the multipath of the first sensor was the largest, so the BER of the first sensor increased like the simulation results in Figure 9. Table 4 demonstrates that the proposed method showed the lowest BER. This result shows that the proposed method can practically overcome the multipath delay.

## 6. Conclusions

This paper proposes a biomimicking modulation method for long-range underwater acoustic communication. The proposed method mimicked the whistle of a large-size whale (i.e., a right humpback whale) and modeled the time-dependent frequency change of the whistles. This model was utilized to overcome the large multipath delay in long-range underwater acoustic communication. Results of the computer simulations and the ocean experiment demonstrate that the BER performance of the proposed method is better than that of conventional studies. In this paper, the novel DoM assessment based on a machine-learning classifier was also utilized to demonstrate that whistles generated by the proposed method are the same as those of the RHW. The assessments showed that the whistles generated with the proposed method were determined as a natural RHW sound, whereas the conventional methods were not. Thus, the proposed method shows better BER and DoM performance than that of conventional methods. 

## Figures and Tables

**Figure 1 sensors-22-08011-f001:**
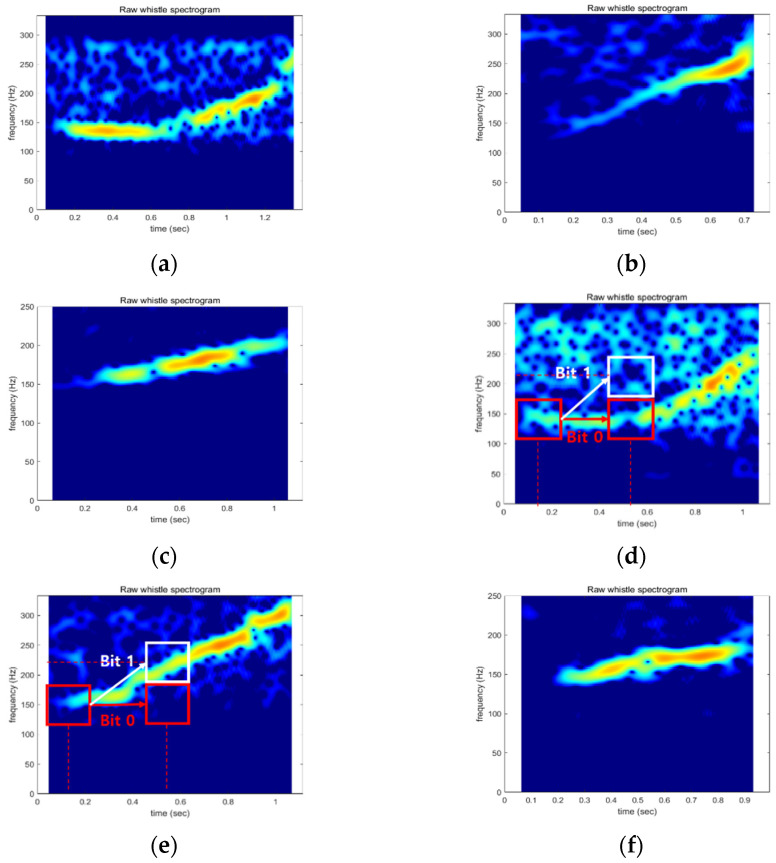
Spectrogram of whistles of the RHW: (**a**), (**d**) Pattern 1; (**b**), (**e**) Pattern 2; (**c**), (**f**) Pattern 3.

**Figure 2 sensors-22-08011-f002:**
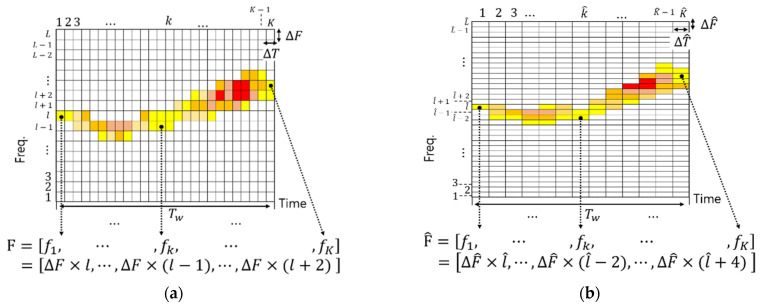
Example of the STFT results of whistles with various ΔT and ΔF: (**a**) STFT with ΔT and (**b**) STFT with $\Delta \hat{T}$.

**Figure 3 sensors-22-08011-f003:**
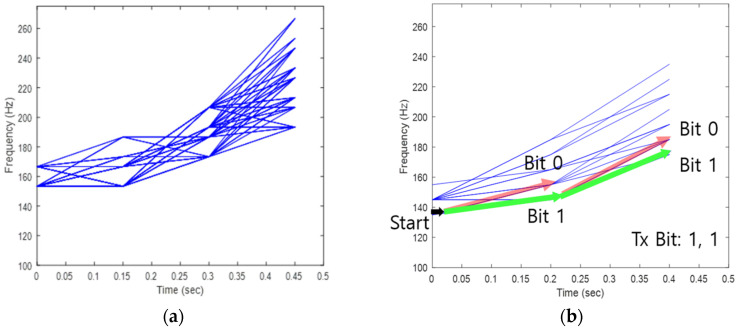
Modeling results of the time-dependent frequency change of the RHW whistle: (**a**) 0.4 s ≤ $T_{w}$ ≤ 0.5 s, ΔT: 0.15 s, ΔF: 13.3 Hz and (**b**) 0.3 s ≤ $T_{w}$ ≤ 0.5 s, ΔT: 0.2 s, ΔF: 10 Hz.

**Figure 4 sensors-22-08011-f004:**
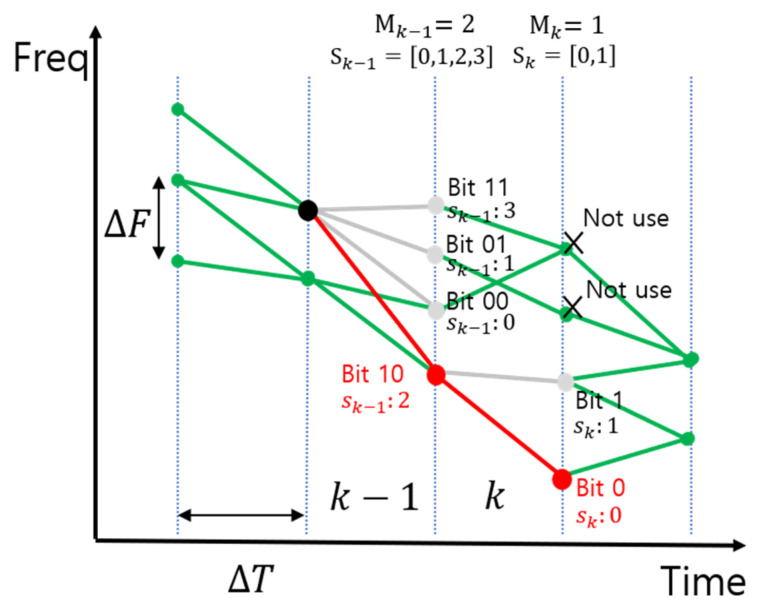
Example of a whistle-mimicking modulation signal.

**Figure 5 sensors-22-08011-f005:**
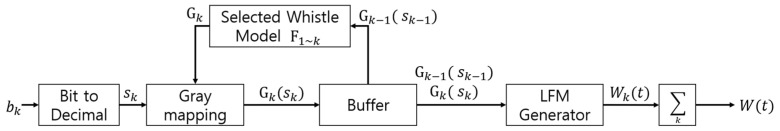
Block diagram of the proposed biomimicking modulation.

**Figure 6 sensors-22-08011-f006:**
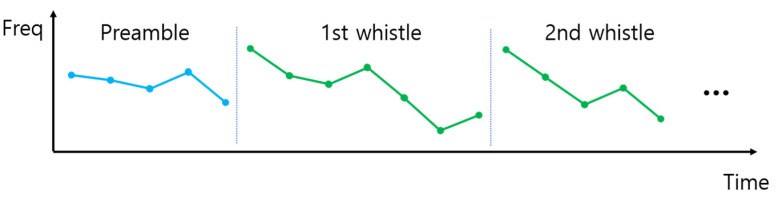
The frame structure of the proposed biomimicking signal.

**Figure 7 sensors-22-08011-f007:**
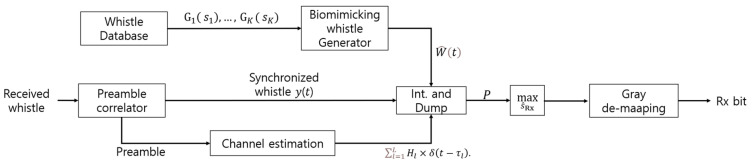
Block diagram of the proposed biomimicking demodulation.

**Figure 8 sensors-22-08011-f008:**
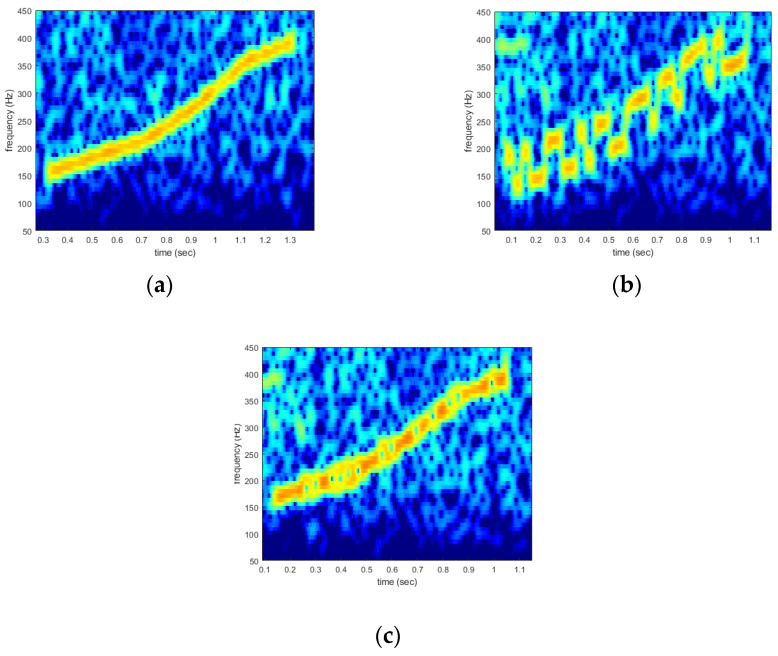
Mimicking whistles for the DoM evaluation: (**a**) the proposed method, (**b**) FSK, and (**c**) CVC-FM and TFSK.

**Figure 9 sensors-22-08011-f009:**
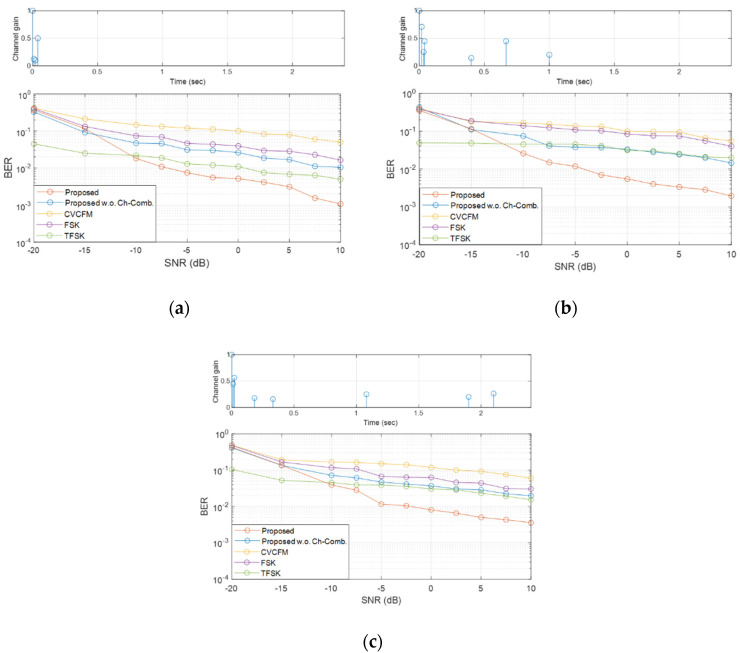
Simulation results of BER performance by delay profile: (**a**) 30 ms channel, (**b**) 1 s channel, and (**c**) 2 s channel.

**Figure 10 sensors-22-08011-f010:**
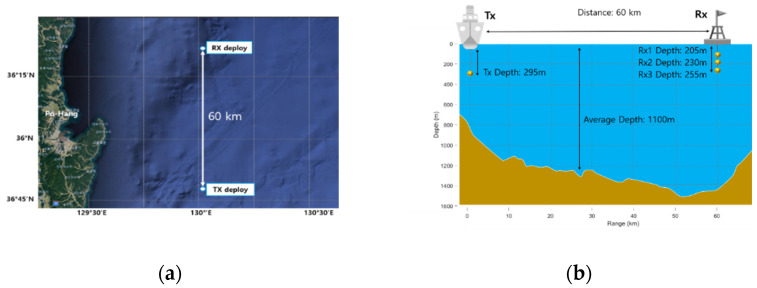
Ocean experiment: (**a**) location, (**b**) configuration.

**Figure 11 sensors-22-08011-f011:**
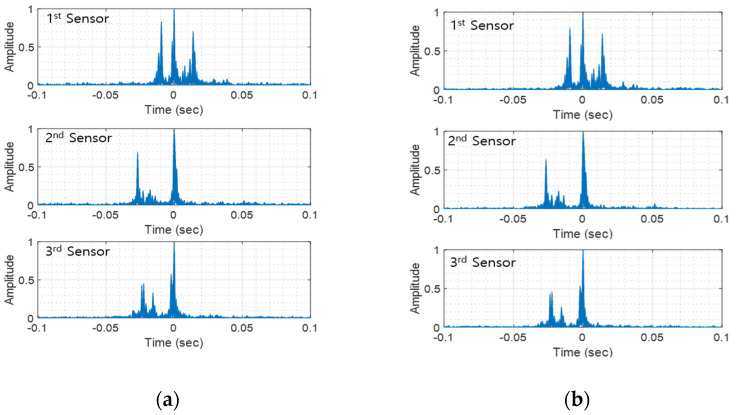
Multipath delay profile of the ocean experiment: (**a**) the proposed method and (**b**) CV-CFM and TFSK.

**Table 1 sensors-22-08011-t001:** The result of the DoM evaluation.

Modulation Scheme	Proposed	FSK	CV-CFM and TFSK
Recognition probability	95%	10%	60%

**Table 2 sensors-22-08011-t002:** Modulation parameters and data rate.

	Proposed						CV-CFM	FSK	TFSK
ΔT (s) symbol duration	0.075	0.1	0.125	0.15	0.175	0.2	0.067	0.0667	0.067 M = 4
ΔF (Hz) bandwidth	26.6	20	16	13.4	11.5	10	15 M = 2	15 M = 2	30 M = 4
Data rate	12 bps						12 bps	15 bps	7 bps

**Table 3 sensors-22-08011-t003:** Summary of long-range acoustic communication experiment and channel.

Ref.	Bandwidth	Range	Number of Paths	Maximum Delay
[16]	Fc: 250 Hz BW: 100 Hz	550 km	2	30 ms
[17]	Fc: 250 Hz BW: 50 Hz	500 km 700 km	1~2	600 ms
[18]	Fc: 500 Hz BW: 100 Hz	100 km	3~4	1 s
[19]	Fc: 400 Hz BW: 100 Hz	300 km	5	1 s
[20]	Fc: 500 Hz BW: 100 Hz	600 km	15~17	2 s
[21]	Fc: 500 Hz BW: 100 Hz	1000 km	5~8	1.5~2.5 s

**Table 4 sensors-22-08011-t004:** BER results of the ocean experiment.

Mod. Scheme and Sensor	Proposed	Proposed w.o. Ch.-Comb.	CV-CFM	TFSK
1st Sensor	0.0094	0.0113	0.0906	0.1394
2nd Sensor	0.0030	0.0057	0.0830	0.1210
3rd Sensor	0.0045	0.0099	0.0635	0.0890
Avg.	0.0056	0.0090	0.0790	0.1165

## Data Availability

Not applicable.

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
