# Peer review of "The Long-Range Biomimetic Covert Communication Method Mimicking Large Whale"

_sensors, 2022, doi:10.3390/s22208011_

Round 1

Reviewer 1 Report

The presented work proposes a novel modulation system for long-distance submarine transmissions. The organization of the contents is clear and well ordered. I consider that the research scheme is correct: characterization of a natural process, model replication proposal for information transmission purposes, simulation of the replicated process and comparison with other similar processes and, finally, field trials to validate the simulation. As a contribution to this study, I would like to suggest to the authors the use of the wavelet transform instead of the STFT, since it allows good temporal resolution for high-frequency events and good frequency resolution for low-frequency events. This is the type of analysis that is preferably used for many real signals. The fixed frequency resolution provided by STFT can make results less accurate from one frequency band to another.

Reviewer 2 Report

The paper discusses a very interesting technique for underwater communications using biologically inspired signals. I like the manuscript, and have several minor remarks only.

First, I recommend reducing the number of mass citations (e.g. [1-14]).  In addition, such an issue appears twice in a first paragraph. 

Second, a brief comparison with other covert communication technologies should be given, e.g. chaos-based communications which utilize noise-like signals. Please, also clarify why pre-recorded whale songs cannot be used and the mimicking model is required to build a communication system.

Third, I recommend reformulating this sentence "...replace the mimicking model of the conventional methods with the large whale.". One can't replace a model with a whale and the benefits of large whale voice over other whales should be emphasized.

More modern approaches for discovering covert communications should be considered, e.g. recurrence quantification and quantified return map analysis. The secrecy of the developed system is discussed very briefly and is not properly evaluated.

Nevertheless, I highly evaluate this comprehensive study (especially the experimental part) and believe that it can be published after only a minor revisions.

Reviewer 3 Report

At present, long-range biomimicking underwater acoustic communication faces two difficulties: first, it needs to overcome high multi-path delay, and second, it needs to maintain high imitation degree under low frequency band. Based on the above problems, this paper proposes a new remote intelligent bionic communication scheme, which aims to achieve lower bit error rate while maintaining high imitation degree in the low-frequency band. This method mainly uses the time-frequency characteristic of whale whistle to make the receiver obtain extra SNR gain from multi-path delay, so as to reduce the BER. Through a large number of theoretical and simulation analysis to demonstrate, the paper also does some experiments.

Some suggestions:

(1) Section 1 mentions that "The whistle-generating habit is that the following frequency values are limited by the frequency value of the previous time. ", which lacks theoretical support. This view is the premise of the design of full text communication scheme, which needs to be supported by clear research conclusions. Please quote relevant literature if available.

(2) In Figure 2, only "∆T is smaller than ∆" is mentioned. Can you give a specific value?

(3) Please explain the lines with different colors in Figure 3, so that readers can understand the modulation mode more intuitively.

(4) The parameter M is not explained in Table 2. Can you point out the calculation formula of the data rate?

(5) All the quotes in the paragraphs below Figure 9 are "In Figure 13", please correct them.

(6) The demonstration of improving multi-path gain in the experiment is very vague, please specify it.

(7) The whole experiment data is too simple, and the data shown in Table 4 cannot reflect its authenticity. It is not enough to prove that the proposed method can obtain a lower bit error rate only by the value of a single one.
